# High Global Health Security Index is a determinant of early detection and response to monkeypox: A cross-sectional study

**Max Carlos Ramírez-Soto** [1,2]*, **Hugo Arroyo-Hernández** [3]

**1** Centro de Investigación en Salud Pública, Facultad de Medicina Humana, Universidad de San Martín de Porres, Lima, Peru, **2** Facultad de Ciencias de la Salud, Universidad Tecnológica del Peru, Lima, Peru, **3** Instituto de Investigaciones en Ciencias Biomédicas, Universidad Ricardo Palma, Lima, Peru

* maxcrs22@gmail.com

## Abstract

### Introduction

Recent outbreaks of monkeypox (Mpox) have occurred in countries outside of Africa, with large numbers of cases spreading rapidly to almost every continent. We aimed to analyze the correlation between the Global Health Security (GHS) Index (categories and indicators) and the Mpox case rate in different regions and globally.

### Methods

In this cross-sectional study, we used data from Mpox cases from the WHO, and the GHS categories and indicators for detection, prevention, reporting, health system, rapid response, international norms compliance, and risk environment. Outcome measures were the relationship between GHS index (categories and indicators) and Mpox case rate using crude and adjusted non-linear regression models.

### Results

After performing adjusted analyses, only risk environment and detection and reporting index were associated with Mpox case rates in the 99 countries and the Region of the Americas, respectively. Antimicrobial resistance (AMR) indicators of the prevention category, risk communication of the rapid response category, the joint external evaluation and performance veterinary services of the norms category, and the infrastructure adequacy of the risk environment category were associated with Mpox case rates in the 99 countries ($p<0.05$). Laboratory systems strength and quality indicator of the detection and reporting category, and emergency response operation indicator of the response rapid category were associated with Mpox case rates in the countries of the region of the Americas ($p<0.05$). AMR indicator of the prevention of the emergence category, and the infrastructure adequacy of the risk environment category were associated with Mpox case rates in the countries of the European Region ($p<0.05$). In the countries of the other regions, only the trade and travel

**Data Availability Statement:** Data used as input into these analyses are publicly available at https://ghsindex.org/archive/ for global health security

indices, and https://worldhealthorg.shinyapps.io/mpx_global/ for Mpox cases.

**Funding:** The author(s) received no specific funding for this work.

**Competing interests:** The authors have declared that no competing interests exist.

restrictions indicator of the rapid response category was associated with Mpox case rates ($p<0.05$).

## Conclusions

Countries, particularly in the Americas region, with high levels of infrastructure adequacy and laboratory system strength and quality as measured by the GHS index are better equipped to detect more Mpox cases. Therefore, they have higher Mpox case detection rates and can successfully respond to Mpox outbreaks.

## 1. Introduction

Since 1970, monkeypox virus (MPXV) infections have been reported in several countries in central and western Africa and in Nigeria [1–4]. Between May 2022 and April 2023, more than 85,000 cases of Mpox were reported in 111 countries that are not endemic for Mpox [5]. Most of these cases have been reported in the Americas and Europe [5, 6]. In Latin America, Peru, Colombia, Chile and Brazil had the highest rates of Mpox per million inhabitants of all ages [7]. As a result, the Mpox outbreak was declared a public health emergency of international concern by the World Health Organization (WHO) in July 2022 [8]. In the current global outbreak, transmission has been associated with sexual contact in high-risk groups [6, 9, 10]. The high incidence of Mpox in the Americas and Europe may be explained by several factors, including the capacity of individual countries to respond, the health system and the ability to detect cases (epidemiological surveillance). Other factors include changes in the behaviour of the population and at-risk groups.

During the COVID-19 pandemic, some studies correlated Global Health Security (GHS) Index scores with SARS-CoV-2 infections, COVID-19 deaths and excess mortality [11–13]. Among the GHS indicators that had associations with COVID-19 mortality rates were access to communication infrastructure, risk environment, government effectiveness, social inclusion and public trust in government [13]. More recently, a report showed that overall GHS index score correlated with Mpox case rates in countries in the Americas (cases per millions) [7]. Another study found that countries with a high overall score on the GHS index had a positive correlation with the number of cases and deaths from Mpox [14]. The GHS indicators are designed to identify gaps in preparedness and encourage countries to strengthen their health security capacities by assessing their ability to detect, prevent and respond to epidemics and pandemics [15]. According to the 2021 GHS Index report, all countries are unprepared for future epidemic/pandemic threats [16]. Despite these correlation analyses, overall GHS scores summarize countries' capacity across all categories. However, countries with high overall scores may still have low scores at the category, indicator and sub-indicator levels that have a more significant impact on outbreak-related outcomes [15]. Because of this, analysis of correlations between country GHS scores and Mpox cases must be performed at the category, indicator and sub-indicator levels, and model weights must be adjusted to reflect the country context when outbreak outcomes are analyzed [15]. In this context, to date, there is no information on the relationship between the Mpox case rate and GHS categories and indicators such as detection, prevention, reporting, health system, rapid response, compliance with international standards, and risk environment at the global and regional levels.

We hypothesized that the GHS categories of detection, prevention, reporting, health system, rapid response, compliance with international standards and risk environment could explain

global and regional Mpox case rates. Thus, we aimed to analyze the correlation between the GHS index (categories and indicators) and Mpox case rates in different countries and regions. The evaluation of the GHS on Mpox is critical for transmission control and will provide important insights into the relationships between the GHS index and Mpox cases rates.

## 2. Methods

### 2.1. Design study

This cross-sectional study was conducted following the guidelines for Strengthening the Reporting of Observational Studies in Epidemiology (STROBE) (**S1 Table**) [17]. This study was an ecological analysis that estimated the association between the rate of global and regional Mpox cases and the GHS index in the current global Mpox outbreak, from May 2022 to April 2023. This is an analysis based on surveillance data reported to WHO by Member States. The data have been submitted by countries in a variety of formats for collation by WHO. Therefore, no ethical approval or participant consent was required to conduct the study.

### 2.2. Data collection

We collected data on the scores from the GHS Index website (https://www.ghsindex.org) [16]. The 2021 GHS Index measures 195 countries' epidemic and pandemic preparedness capacity. According to this information, all countries continue to be unprepared for the threat of epidemics and pandemics in the future. The 2021 GHS Index comprises 171 questions grouped into 37 indicators across six overarching categories. These categories include the overall score, prevention, detection and reporting, rapid response, health systems, standards and the risk environment. In this study, we have included the six categories of the GHS index for countries from seven regions that have reported cases of Mpox. In addition, we have included indicators that introduce each category to provide a better explanation of the relationship between the GHS index and Mpox cases (**S2 Table**). The overall score (0 to 100) for each country is a weighted sum of the scores in the six categories. The score for each category is given on a scale from 0 to 100. A score of 100 represents the most favourable health security conditions, but does not mean that a country has a perfect national health security situation. A score of 0 is the least favourable condition, but does not imply a lack of capacity [16]. Each category is standardized based on of the sum of its underlying indicators and sub-indicators, and a weight is assigned to each. The weights in the model are dynamic and can be changed by the users. To facilitate reproducible comparisons across countries, categories and indicators, measurements have been normalized to a scale of 0 to 100 [16].

In May 2022, the WHO established a global surveillance system. This has been extended retroactively to 1 January 2022 and beyond. Mpox cases were extracted from the World Health Organization, 2022 Mpox Outbreak: Global Trends website (https://worldhealthorg.shinyapps.io/mpx_global/) [5]. This website provides a global overview of the Mpox epidemiological situation as reported to WHO as of April 02, 2023. The report includes laboratory-confirmed cases as defined by the WHO's working case definition published in the Surveillance, case investigation, and contact tracing for Mpox interim guidance. Countries with 0 cases by 04 April 2023 were excluded from the analysis. Countries with Mpox cases where the GHS index score was also not available were excluded from the study.

Population data of each country was obtained from the website of "The World Bank" (https://data.worldbank.org/indicator/SP.POP.TOTL), updated as of September 31, 2022 [18]. Total population refers to a de facto definition of population that includes all residents regardless of legal status or nationality.

## 2.3. Statistical analysis

We estimated the cumulative crude rates of Mpox in each country (per 1,000,000 inhabitants). The cumulative rates of Mpox were calculated by dividing the number of Mpox cases per country, by the estimated population of each country. Population counts used to calculate the cumulative case rates were obtained from projections from The World Bank. Means and standard deviation (SD) or median and interquartile range (IQR) were calculated for Mpox case rates, GHS Index score, and the categories, prevention of the emergency, detection and reporting, respond rapid, health systems, norms, and risk environment. Scatter plots were also provided to indicate the correlation between the GHS Index and its six indicators, and Mpox case rate for all the countries included, Region of the Americas, European region, and other regions.

We also assessed the association between the increase in Mpox cases rate, and the six GHS indexes in all the countries included, the region of the Americas region, the European region, and other regions. Subsequently, non-linear regression analyses were performed with the Kernel-based regularized least squares (KRLS) method to explore the association between an increase in Mpox cases rate, the GHS index score, the categories, prevention of the emergency, detection and reporting, respond rapid, health systems, norms, and risk environment, and the indicators of each one of these six categories. These estimates were reported as an estimate of the average point marginal effect (Ave) and its standard error (ES). Calculations have been made for all countries included in the Americas, Europe and Other regions. In addition, adjusted analyses were performed to assess the association between Mpox case rate and the GHS index. In adjusted analyses, forward stepwise selection was used to select statistically significant regressors ($p < 0.05$). The adjusted coefficient of determination (R2) shows the probability of the independent variables explaining the dependent variable in multiple regression models. P-values $<0.05$ were considered significant. Statistical analyses were done using StataSE 17.0 Software.

## 3. Results

### 3.1. Descriptive statistics

From 01 June 2022 to 04 April 2023, a total of 86,601 Mpox cases were reported in 99 countries (24 countries in the region of the Americas, 43 countries in the Europe region and 32 countries in other regions). The crude cumulative case rate of Mpox was higher in the Americas (22.8, IQR 4.8; 45.9) and Europe (16.2, IQR 3.2; 50.6). The average GHS score was higher in the countries of the European Region (52.8 SD 9.9) (**Table 1**).

**Fig 1** shows the rate of Mpox cases per million habitants, and the GHS index indicators for the total score and the six indicators for each country. In the Region of the Americas, the cumulative case rate was highest in Peru, followed by the USA, Colombia, Chile and Brazil. In the European region, the crude cumulative case rate was highest in Spain, followed by Portugal, Luxembourg, Micronesia and Belgium (**Fig 1**). In the other regions, the crude cumulative case rates were low (**Fig 1**).

### 3.2. Regression analysis for the overall GHS index score and its six indicators

Country and region-level correlations between the GHS Index categories and Mpox case rates are shown in **Table 2**. GHS Index score, incidence prevention, detection and notification, rapid response, health systems and risk environment were associated with Mpox case rates in the 99 countries (**Fig 2** and **Table 2**). In the countries of the region of the Americas, GHS

**Table 1. Mpox case rates and mean of Global Health Security indicators.**

| | Mpox case rates | Global Health Security Index Score | Average score of the Global Health Security categories | | | | | |
| --- | --- | --- | --- | --- | --- | --- | --- | --- |
| | Average (IQR) | | Prevention of the emergence | Detection and reporting | Respond rapid | Health systems | Norms | Risk environment |
| All regions | 5.3 (0.7; 31.1)* | 46.8 (13.2) | 38.3 (16.9) | 42.0 (19.7) | 41.8 (34.0; 51.8)* | 46.5 (24.0; 54.7)* | 52.7 (14.2) | 62.4 (14.5) |
| Region of the Americas | 22.8 (4.8; 45.9)* | 43.3 (14.2) | 35.3 (18.9) | 37.1 (21.9) | 38.6 (34.6; 49.5)* | 37.3 (18.3; 54.8)* | 49.4 (14.0) | 57.3 (10.8) |
| European Region | 16.2 (3.2; 50.6)* | 52.8 (9.9) | 46.6 (13.1) | 45.1 (17.2) | 45.5 (10.6) | 51.2 (12.3) | 59.7 (51.4; 63.9)* | 73.4 (62.9; 79.9)* |
| Other regions | 0.4 (0.1; 2.9)* | 41.6 (13.5) | 29.3 (15.8) | 41.6 (21.1) | 36.9 (29.9; 51.2) | 33.8 (17.1) | 49.4 (14.2) | 54.8 (14.6) |

*50th percentile (25th percentile; 75th percentile), rest of values are median and standard deviation. IQR, Interquartile range.

Index score, prevention, detection and notification, respond rapid, and norms were also associated with Mpox case rates (**Fig 3** and **Table 2**). In the countries of European region, only the risk environment score was associated with Mpox case rates (**Fig 4** and **Table 2**). In the countries of the other regions, GHS Index and its scores were not associated with Mpox case rates (**Fig 5** and **Table 2**).

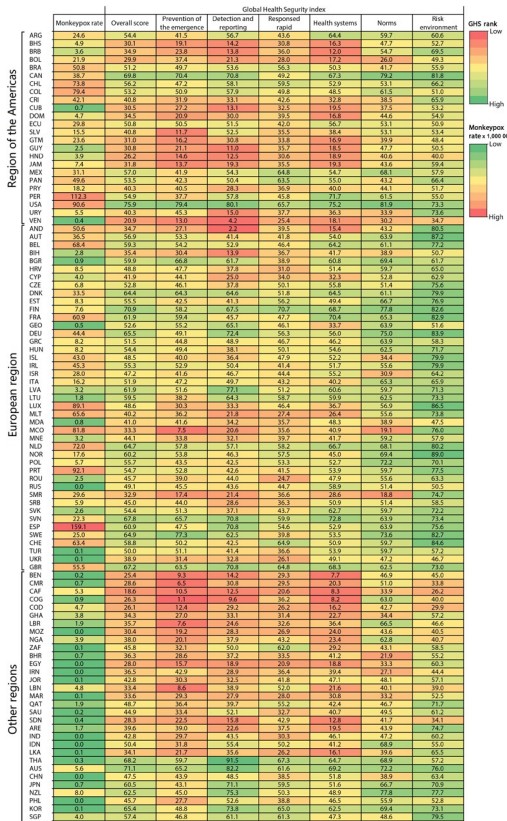

**Fig 1. Mpox cases per million habitants and GHS index indicators for total score and six indicators for countries and regions.**

**Table 2. Crude non-linear regression analysis between the Global Health Security Index categories and Mpox case rates.**

| | Average† | SE | p-value |
|---|---|---|---|
| **All regions (99 countries)** | | | |
| **Global Health Security Index score** | 0.42 | 0.12 | **0.001** |
| Prevention of the emergence score | 0.27 | 0.08 | **0.003** |
| Detection and reporting score | 0.16 | 0.05 | **0.005** |
| Respond rapid score | 0.51 | 0.14 | **0.001** |
| Health systems score | 0.42 | 0.12 | **0.001** |
| Norms score | 0.17 | 0.12 | 0.17 |
| Risk environment score | 0.64 | 0.16 | **<0.001** |
| **Region of the Americas (24 countries)** | | | |
| **Global Health Security Index score** | 0.77 | 0.21 | **0.001** |
| Prevention of the emergence score | 0.42 | 0.14 | **0.008** |
| Detection and reporting score | 0.56 | 0.14 | **0.001** |
| Respond rapid score | 1.24 | 0.33 | **0.001** |
| Health systems score | 0.87 | 0.54 | 0.123 |
| Norms score | 0.5 | 0.22 | **0.03** |
| Risk environment score | 0.19 | 0.14 | 0.199 |
| **European Region (43 countries)** | | | |
| **Global Health Security Index score** | 0.2 | 0.13 | 0.127 |
| Prevention of the emergence score | 0.12 | 0.11 | 0.285 |
| Detection and reporting score | 0.05 | 0.07 | 0.468 |
| Respond rapid score | 0.15 | 0.11 | 0.179 |
| Health systems score | 0.05 | 0.11 | 0.683 |
| Norms score | 0.05 | 0.14 | 0.708 |
| Risk environment score | 0.95 | 0.35 | **0.01** |
| **Other regions (32 countries)** | | | |
| **Global Health Security Index score** | -0.005 | 0.007 | 0.502 |
| Prevention of the emergence score | -0.0002 | 0.01 | 0.974 |
| Detection and reporting score | 0.0002 | 0 | 0.959 |
| Respond rapid score | -0.0004 | 0.01 | 0.956 |
| Health systems score | -0.00074 | 0 | 0.866 |
| Norms score | 0.0017 | 0.01 | 0.781 |
| Risk environment score | 0.027 | 0.02 | 0.252 |

†Kernel-based regularized least squares (KRLS) are used and the average of the point marginal effects are reported.

SE: standard error. Dependent variable is Mpox case rate in each country (per 1,000 000 inhabitants).

After performing adjusted analyses, only risk environment (Ave 0.24, SE 0.09; p value = 0.005, R2 = 0.23) and detection and reporting score (Ave 0.23, SE 0.08, p value = 0.006; R2 = 0.60) were associated with Mpox case rates in the 99 countries and the region of the Americas, respectively (**Table 3**).

## 3.3. Regression analysis for the indicators of the six GHS index score

Country and region-level crude non-linear regression analysis between the GHS Index indicators and Mpox case rates are shown in **Table 4**. Adjusted analyses were performed on statistically significant results.

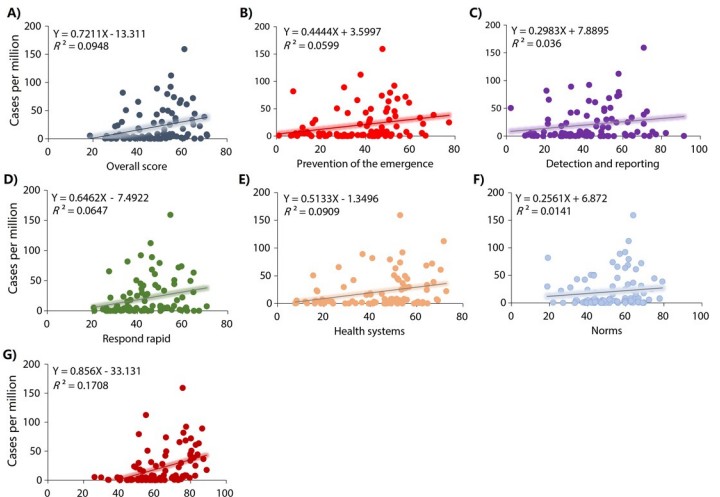

**Fig 2. Relationships between the GHS index categories and Mpox cases per millions habitants in 99 countries.** A) Overall score. B) Prevention: Prevention of the emergence or release of pathogens. C) Detection and reporting: Early detection and reporting for epidemics of potential int'l concern. D) Respond rapid: Rapid response to and mitigation of the spread of an epidemic. E) Health system: sufficient & robust health sector to treat the sick & protect health workers. F) Norms: Commitments to improving national capacity, financing and adherence to norms. G) Risk environment: Overall risk environment and country vulnerability to biological threats.

Country and region-level adjusted non-linear regression analysis between the GHS Index indicators and Mpox case rates are shown in **Table 5**. In adjusted model 3, antimicrobial resistance (AMR) indicator (Ave 0.19, SE = 0.07, p = 0.008) of the prevention of the emergence category, risk communication (Ave 0.19, SE 0.09, p = 0.035) of the rapid response category, the joint external evaluation (JEE) and performance veterinary services (PVS) (Ave -0.31, SE = 0.09, p = 0.002) of the norms category, and the adequacy of infrastructure (Ave 0.24, SE = 0.08, p = 0.003) of the risk environment category were associated with Mpox cases rates in the 99 countries (R2 of 0.43) (**Table 5**). Laboratory systems strength and quality indicator (Ave 0.33, SE = 0.11, p = 0.008) of the detection and reporting category, and emergency response operation indicator (Ave 0.26, SE = 0.09, p = 0.007) of the response rapid category were associated with Mpox cases rates in the countries of the Region of the Americas (R2 of 0.80) (**Table 5**). AMR (Ave 0.33, SE = 0.13, p = 0.016) of the prevention of the emergence category, and the infrastructure adequacy (Ave 0.39, SE = 0.15, p = 0.012) of the risk environment category were associated with Mpox cases rates in the European region countries (R2 of 0, 43) (**Table 5**). In the countries of the other regions, risk communication and trade and travel restrictions of the rapid response category, access to healthcare of the health systems category, and political and security risk of the risk environment category were associated with Mpox case rates in adjusted model 2; however, only trade and travel restrictions of the rapid response category was associated with Mpox case rates in adjusted model 3 (Ave 0.01, SE<0.01, p = 0.048) (**Table 5**).

## 4. Discussion

This analysis is the first to provide a direct comparison between Mpox case rates and the GHS Index categories and indicators in different countries and regions. There are multiple factors for understanding the impact of Mpox outbreak relative to existing public health assessments. These factors include lack and availability of high-quality data, changes between populations, lack of resources, public health indicators available and vaccination against Mpox data.

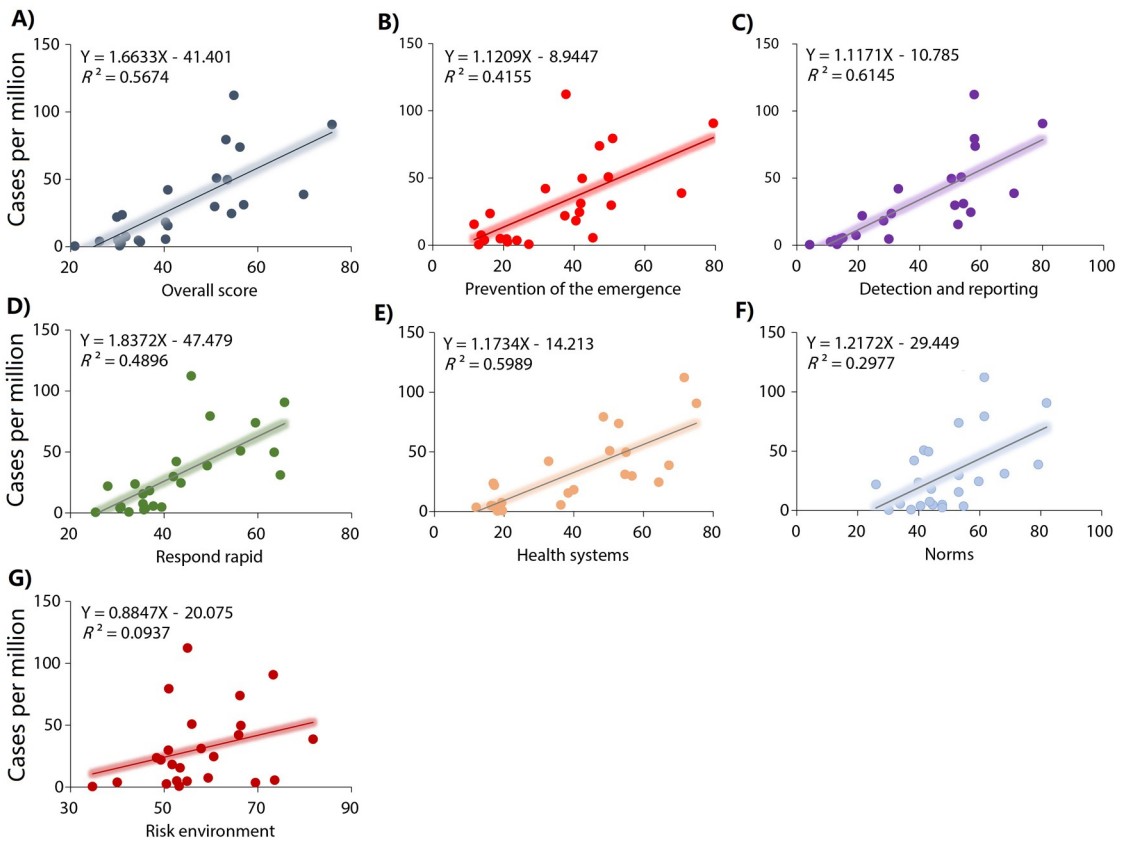

**Fig 3. Relationships between the GHS index categories and Mpox cases per millions habitants in the region of the Americas.**
A) Overall score. B) Prevention: Prevention of the emergence or release of pathogens. C) Detection and reporting: Early detection and reporting for epidemics of potential int'l concern. D) Respond rapid: Rapid response to and mitigation of the spread of an epidemic. E) Health system: sufficient & robust health sector to treat the sick & protect health workers. F) Norms: Commitments to improving national capacity, financing and adherence to norms. G) Risk environment: Overall risk environment and country vulnerability to biological threats.

Currently, Mpox cases information in WHO is only available for 114 countries [5]. Case detection and notification are affected by laboratory capacity, infrastructure, surveillance epidemiological, and other factors. Therefore, middle-income and high-income countries are likely to have a more robust health security capacity for outbreak and emergency response. In this context, the GHS Index aims to identify gaps and opportunities for strengthening health systems. While the results of the GHS Index show that no country is prepared to respond to an epidemic or pandemic [4, 16], our findings suggest that countries with robust capacities for preparedness, response and effective risk communication can detect more Mpox cases and respond better to health emergencies. However, it is also important to note that one study reported that national Gross Domestic Product (GDP) and GDP per capita were weakly correlated with the overall GHS index. Therefore, the categories and indicators in the GHS Index may be biased towards middle- and high-income countries in the region of Americas and Europe region, where high rates of Mpox cases have been reported [19].

We found a relationship between countries with better conditions for managing a risk environment and vulnerability to biological threats, and Mpox case rates after performing adjusted analyses. This category includes socio-economic, political, regulatory and ecological factors that increase the vulnerability to an outbreak [20]. In this category, infrastructure improvements were associated with Mpox case rates in the region of the Americas and Europe region.

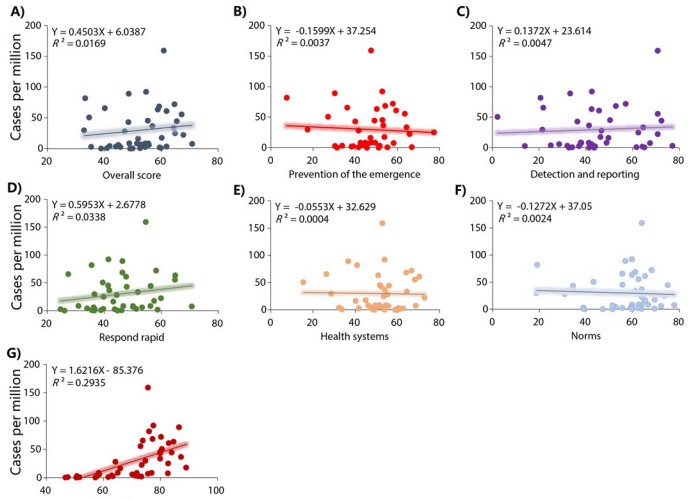

**Fig 4. Relationships between the GHS index categories and Mpox cases per millions habitants in European region.** A) Overall score. B) Prevention: Prevention of the emergence or release of pathogens. C) Detection and reporting: Early detection and reporting for epidemics of potential int'l concern. D) Respond rapid: Rapid response to and mitigation of the spread of an epidemic. E) Health system: sufficient & robust health sector to treat the sick & protect health workers. F) Norms: Commitments to improving national capacity, financing and adherence to norms. G) Risk environment: Overall risk environment and country vulnerability to biological threats.

Approximately 23% of countries have high GHS Index scores, including the risk environment [16]. Among these countries, the USA, Spain or Belgium have more effective surveillance systems to detect emergencies, according to previous studies [11, 21]. In contrast, a study showed that a higher level of health safety capacity in the GHS Index, is associated with lower excess COVID-19 mortality [13, 22]. This relationship with improved infrastructure may have been essential for the implementation of risk communication strategies in the community, and therefore for the detection of more Mpox cases. Thus, a solid health infrastructure and preventive measures are critical to responding to a health crisis and detecting an outbreak.

The GHS provides valuable data to measure the adequacy of infrastructure or the strength and quality of laboratory systems in low, middle and high income countries at the time of the spread of aetiological agents with pandemic potential. However, if there is a large degree of dispersion in these data, the presence of outliers may affect the correlation coefficient and hence the results of the correlation. In this context, our findings also show that the early detection and reporting category and its indicators of laboratory system strength and quality, designed to assess laboratory capacity to detect priority diseases, were associated with Mpox case rates in the Americas after performing adjusted analyses. This finding may be related to the capacity for reliable and timely case detection, particularly in countries with stronger health surveillance systems for public health emergencies [11]. Although these variables are not unique to successful epidemic/pandemic containment, they are correlated according to their magnitude in a country or region. On the other hand, only the AMR indicator in the prevention of the emergence or release of pathogens category was associated with Mpox case rates in the overall analysis and in the European Region. This indicator may be related to high-income countries, as documented in a correlation analysis between GHSI-AMR and GDP per capita for high-income countries and those with high investment in health [23]. Lastly, the risk communication indicator in the rapid response category was correlated with an increase in the rate of Mpox cases. This suggests that countries with a better public health emergency

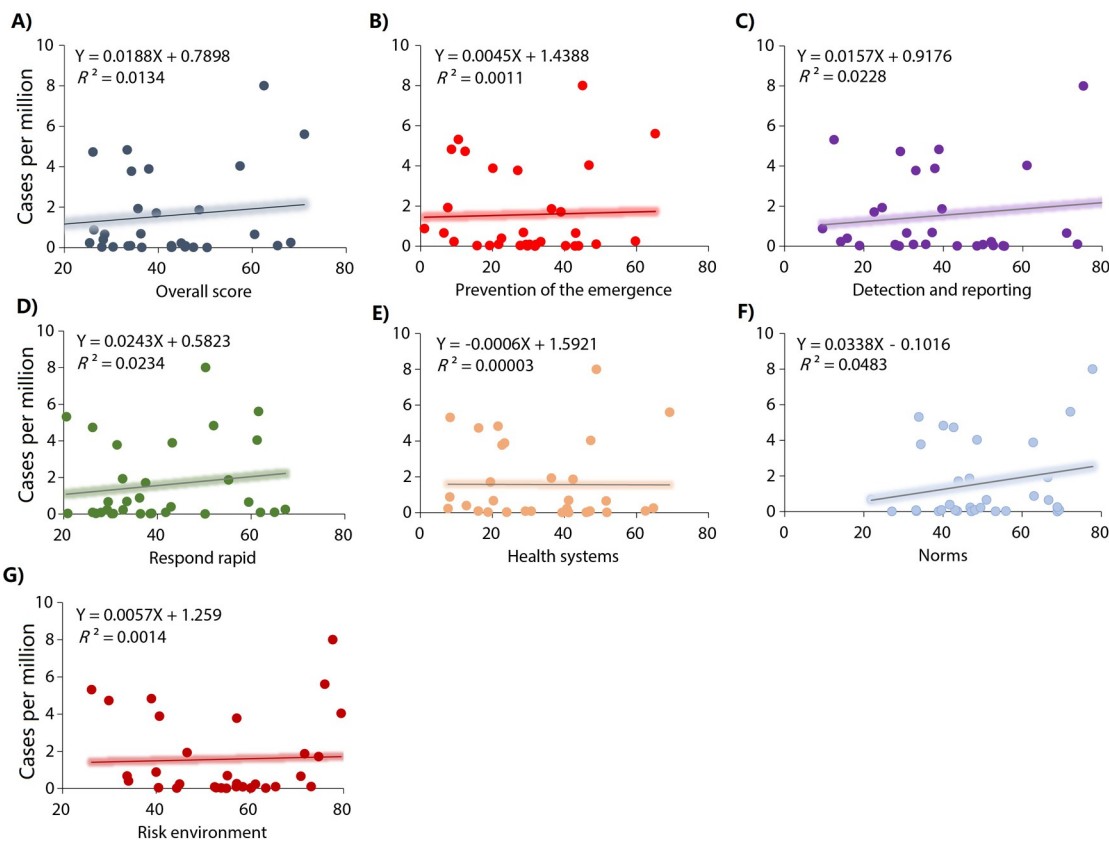

**Fig 5. Relationships between the GHS index categories and Mpox cases per millions habitants in others regions.** A) Overall score. B) Prevention: Prevention of the emergence or release of pathogens. C) Detection and reporting: Early detection and reporting for epidemics of potential int'l concern. D) Respond rapid: Rapid response to and mitigation of the spread of an epidemic. E) Health system: sufficient & robust health sector to treat the sick & protect health workers. F) Norms: Commitments to improving national capacity, financing and adherence to norms. G) Risk environment: Overall risk environment and country vulnerability to biological threats.

**Table 3. Adjusted non-linear regression analysis between the Global Health Security Index categories and Mpox case rates.**

|  | Adjusted analysis | | | |
| --- | --- | --- | --- | --- |
|  | Average† | SE | p-value | R2 |
| **All regions (99 countries)** |  |  |  |  |
| Prevention of the emergence score | 0.04 | 0.07 | 0.579 | 0.23 |
| Detection and reporting score | 0.02 | 0.06 | 0.73 |  |
| Respond rapid score | 0.1 | 0.1 | 0.331 |  |
| Health systems score | 0.11 | 0.06 | 0.086 |  |
| Risk environment score | 0.24 | 0.09 | **0.005** |  |
| **Region of the Americas (24 countries)** |  |  |  |  |
| Prevention of the emergence score | 0.09 | 0.11 | 0.42 | 0.6 |
| Detection and reporting score | 0.23 | 0.08 | **0.006** |  |
| Respond rapid score | 0.28 | 0.14 | 0.057 |  |
| Norms score | 0.07 | 0.14 | 0.648 |  |

†Kernel-based regularized least squares (KRLS) are used and the average of the point marginal effects are reported. Dependent variable is Mpox case rate in each country (per 1,000 000 inhabitants).

**Table 4. Crude non-linear regression analysis between the Global Health Security Index indicators and Mpox case rates.**

| Indicators by category | All regions | | | Region of the Americas | | | European Region | | | Other regions | | |
|---|---|---|---|---|---|---|---|---|---|---|---|---|
| | Average† | SE | p-value | Average† | SE | p-value | Average† | SE | p-value | Average† | SE | p-value |
| **Prevention of the emergence score** | | | | | | | | | | | | |
| Antimicrobial resistance (AMR) | 0.47 | 0.1 | **<0.001** | 0.27 | 0.12 | **0.035** | 0.46 | 0.13 | **0.001** | <0.01 | <0.01 | 0.228 |
| Zoonotic disease | 0.1 | 0.05 | **0.037** | 0.65 | 0.2 | **0.003** | -0.003 | 0.08 | 0.967 | <0.01 | 0.01 | 0.984 |
| Biosecurity | 0.15 | 0.06 | **0.006** | 0.37 | 0.11 | **0.003** | 0.05 | 0.08 | 0.571 | <0.01 | <0.01 | 0.85 |
| Biosafety | 0.07 | 0.02 | **0.006** | 0.2 | 0.05 | **0.001** | -0.03 | 0.03 | 0.356 | <0.01 | <0.01 | 0.819 |
| Dual-use research and culture of responsible science | <0.01 | 0.01 | 0.9 | 0.01 | 0.01 | 0.334 | -0.01 | 0.01 | 0.225 | <0.01 | <0.01 | 0.081 |
| Immunization | <0.01 | 0.03 | 0.922 | 0.02 | 0.06 | 0.759 | -0.02 | 0.07 | 0.744 | -0.002 | <0.01 | 0.461 |
| **Detection and reporting score** | | | | | | | | | | | | |
| Laboratory systems strength and quality | 0.1 | 0.06 | 0.106 | 0.36 | 0.14 | **0.015** | 0.04 | 0.07 | 0.58 | <0.01 | <0.01 | 0.819 |
| Laboratory supply chains | 0.02 | 0.01 | 0.102 | NC | | | 0.01 | 0.01 | 0.394 | <0.01 | <0.01 | 0.476 |
| Real-time surveillance and Reporting | 0.1 | 0.04 | **0.011** | 0.32 | 0.09 | **0.001** | 0.03 | 0.04 | 0.374 | <0.01 | <0.01 | 0.224 |
| Surveillance data accessibility and transparency | 0.31 | 0.08 | **<0.001** | 0.21 | 0.07 | **0.006** | 0.05 | 0.04 | 0.21 | <0.01 | <0.01 | 0.848 |
| Case-based investigation | <0.01 | 0.03 | 0.975 | 0.12 | 0.05 | **0.041** | -0.06 | 0.05 | 0.19 | <0.01 | <0.01 | 0.895 |
| Epidemiology workforce | 0.04 | 0.04 | 0.258 | 0.07 | 0.04 | 0.088 | 0.45 | 0.38 | 0.243 | -0.001 | <0.01 | 0.804 |
| **Respond rapid score** | | | | | | | | | | | | |
| Emergency preparedness and response planning | 0.04 | 0.03 | 0.149 | 0.25 | 0.1 | **0.021** | -0.08 | 0.06 | 0.133 | <0.01 | <0.01 | 0.932 |
| Exercising response plans | -0.02 | 0.06 | 0.772 | NC | | | -0.21 | 0.5 | 0.684 | 0.01 | 0.01 | 0.275 |
| Emergency response operation | -0.001 | 0.01 | 0.878 | 0.05 | 0.02 | **0.007** | -0.01 | 0.02 | 0.612 | -0.003 | <0.01 | 0.113 |
| Linking public health and security authorities | NC | | | NC | | | NC | | | NC | | |
| Risk communication | 0.16 | 0.06 | 0.007 | 0.11 | 0.06 | 0.083 | 0.13 | 0.06 | 0.055 | 0.01 | <0.01 | **0.039** |
| Access to communications Infrastructure | 0.47 | 0.28 | 0.099 | 0.28 | 0.14 | 0.068 | 0.07 | 0.19 | 0.733 | 0.01 | <0.01 | 0.241 |
| Trade and travel restrictions | 0.05 | 0.12 | 0.651 | -0.17 | 0.09 | 0.071 | 0.06 | 0.08 | 0.401 | 0.02 | 0.01 | **0.009** |
| **Health systems score** | | | | | | | | | | | | |
| Health capacity in clinics, hospitals and community care centers | 0.25 | 0.08 | **0.004** | 0.34 | 0.13 | **0.016** | 0.1 | 0.09 | 0.297 | <0.01 | <0.01 | 0.596 |
| Supply chain for health system and healthcare workers | 0.08 | 0.05 | 0.09 | 0.96 | 0.28 | **0.002** | -0.03 | 0.08 | 0.722 | <0.01 | <0.01 | 0.976 |
| Medical countermeasures and personnel deployment | 0.01 | 0.01 | 0.22 | 0.17 | 0.05 | **0.004** | -0.01 | 0.02 | 0.38 | NC | | |
| Healthcare access | -0.19 | 0.15 | 0.212 | 0.23 | 0.33 | 0.49 | -0.39 | 0.23 | 0.099 | -0.13 | 0.05 | **0.017** |
| Communications with healthcare workers during a public health emergency | 0.02 | 0.01 | 0.153 | 0.12 | 0.07 | 0.08 | 0.02 | 0.02 | 0.394 | <0.01 | <0.01 | 0.393 |
| Infection control practices | NC | | | NC | | | NC | | | NC | | |
| Capacity to test and approve new medical countermeasures | 0.01 | <0.01 | **0.071** | 0.52 | 0.1 | **<0.001** | 0.44 | 0.19 | **0.023** | <0.01 | <0.01 | 0.536 |
| **Norms score** | | | | | | | | | | | | |
| IHR reporting compliance and disaster risk reduction | 0.28 | 0.09 | **0.003** | NC | | | <0.01 | 0.02 | 0.831 | 0.001 | <0.01 | 0.23 |
| Cross-border agreements on public health and animal health emergency response | 0.07 | 0.03 | **0.004** | 0.02 | 0.02 | 0.333 | 0.02 | 0.02 | 0.495 | <0.01 | <0.01 | 0.332 |
| International commitments | 0.05 | 0.06 | 0.426 | -0.66 | 0.56 | 0.249 | -0.03 | 0.06 | 0.546 | <0.01 | <0.01 | 0.506 |
| JEE (Joint External Evaluation) and PVS (Performance Veterinary Services) | -0.15 | 0.05 | **0.004** | -0.09 | 0.22 | 0.675 | NC | | | <0.01 | <0.01 | 0.662 |
| Financing | -0.03 | 0.03 | 0.362 | 0.03 | 0.06 | 0.641 | -0.11 | 0.11 | 0.329 | -0.001 | <0.01 | 0.807 |
| Commi5ment to sharing of genetic & biological data & specimens | <0.01 | <0.01 | 0.071 | NC | | | NC | | | NC | | |
| **Risk environment score** | | | | | | | | | | | | |
| Political and security risk | 0.27 | 0.09 | **0.003** | -0.03 | 0.09 | 0.734 | 0.23 | 0.29 | 0.441 | 0.02 | 0.01 | **0.021** |

*(Continued)*

**Table 4.** (Continued)

| Indicators by category | All regions | | | Region of the Americas | | | European Region | | | Other regions | | |
|---|---|---|---|---|---|---|---|---|---|---|---|---|
| | Average† | SE | p-value | Average† | SE | p-value | Average† | SE | p-value | Average† | SE | p-value |
| Socio-economic resilience | 0.36 | 0.11 | **0.001** | 0.08 | 0.11 | 0.5 | 0.42 | 0.19 | **0.033** | <0.01 | <0.01 | 0.654 |
| Infrastructure adequacy | 0.52 | 0.1 | **<0.001** | 0.33 | 0.11 | **0.008** | 0.66 | 0.19 | **0.001** | <0.01 | 0.01 | 0.868 |
| Environmental risks | 0.08 | 0.07 | 0.251 | 0.06 | 0.11 | 0.564 | 0.11 | 0.15 | 0.464 | <0.01 | 0.01 | 0.709 |
| Public health vulnerabilities | 0.49 | 0.15 | **0.002** | 0.13 | 0.26 | 0.616 | 0.43 | 0.16 | **0.013** | 0.01 | 0.01 | 0.346 |

†Kernel-based regularized least squares (KRLS) are used and the average of the point marginal effects. Dependent variable is Mpox case rate in each country (per 1,000 000 inhabitants). If two or more indicators were significant in a category, they entered the adjusted model 1 and if only one indicator was significant, they entered the adjusted model 2.

Abbreviations: NC; no convergent.

communication strategy may be able to detect more cases, which would be associated with having a strong health infrastructure and better health services [13].

In the norms category, the JEE and PVS indicators were negatively associated with Mpox case rates in the overall analysis for all countries. This indicator is a collaborative, voluntary and externally validated assessment of countries' preparedness for, detection of and rapid response to public health threats. This assesses commitments to improve national capacities, funding and alignment with global standards on international health rules. Notably, only 44 of the 99 countries surveyed had information [16], and most of these were countries with low case rates of Mpox. Therefore, it is likely that these 44 countries that were analyzed have shown a greater commitment to international standards for the response to public health emergencies [12, 24, 25].

For the countries in the other regions, only the trade and travel restriction indicator in the rapid response category was associated with Mpox case rates. A possible explanation is that by observing an increase in Mpox case rates, these countries may have improved early detection and reporting of cases, as these indicators have a greater impact on the GHS index, even though they may not limit the spread of Mpox [26]. However, since underreporting of Mpox cases is a common problem in countries with low GHS Index scores, these places may not have reliable data, so Mpox case rates are underestimated. Therefore, countries with fewer human resources for health and low GHS Index scores are more vulnerable to epidemics and pandemics due to limitations in essential health services. It is also important to note that a differentiated analysis was not possible due to the limited data available for these countries in the proposed models. Therefore, the directionality of correlations should be interpreted cautiously.

## 4.1. Limitations and strengths

Our study has several limitations. First, any measurement of countries' performance on the GHS Index is subject to the nature of the outbreak and the lack of primary information on Mpox cases. Though GHS Index indicators can explain Mpox case rates, countries vary in their ability to detect and report Mpox cases. Consequently, the results of this study may change as the outbreak evolves and more information becomes available. Second, the association with the GHS Index may be limited by the lack of robust surveillance capacity in some countries, which may not be able to adequately track and report cases. For example, only 44 countries had JEE and PVS indicators available in the norms category. Therefore, this may have contributed to biased estimates. Third, our findings are also subject to the limitations of

**Table 5. Adjusted nonlinear regression analysis between the Global Health Security Index indicators and Mpox case rates.**

| Indicators by category | Model 1* | | | | Model 2** | | | | Model 3*** | | | |
|---|---|---|---|---|---|---|---|---|---|---|---|---|
| | Average† | SE | p-value | R2 | Average† | SE | p-value | R2 | Average† | SE | p-value | R2 |
| **All regions (99 countries)** | | | | | | | | 0.45 | | | | 0.43 |
| **Prevention of the emergence Score** | | | | | | | | | | | | |
| Antimicrobial resistance (AMR) | 0.29 | 0.09 | 0.001 | 0.29 | 0.12 | 0.06 | **0.04** | | 0.19 | 0.07 | **0.008** | |
| Zoonotic disease | 0.01 | 0.13 | 0.92 | | | | | | | | | |
| Biosecurity | 0.07 | 0.1 | 0.509 | | | | | | | | | |
| Biosafety | 0.04 | 0.71 | 0.479 | | | | | | | | | |
| **Detection and reporting score** | | | | | | | | | | | | |
| Real-time surveillance and Reporting | 0.04 | 0.06 | 0.453 | 0.19 | | | | | | | | |
| Surveillance data accessibility and transparency | 0.19 | 0.06 | 0.002 | | 0.07 | 0.05 | 0.191 | | | | | |
| **Respond rapid score** | | | | | | | | | | | | |
| Risk communication | | | | | 0.16 | 0.08 | **0.04** | | 0.19 | 0.09 | **0.035** | |
| **Health systems score** | | | | | | | | | | | | |
| Health capacity in clinics, hospitals and community care centers | | | | | -0.03 | 0.08 | 0.727 | | | | | |
| **Norms score** | | | | | | | | | | | | |
| Cross-border agreements on public health and animal health emergency response | 0.08 | 0.04 | 0.036 | 0.19 | 0.04 | 0.04 | 0.325 | | | | | |
| JEE (Joint External Evaluation) and PVS (Performance Veterinary Services) | -0.19 | 0.09 | 0.029 | | -0.21 | 0.08 | **0.013** | | -0.31 | 0.09 | **0.002** | |
| **Risk environment score** | | | | | | | | | | | | |
| Political and security risk | 0.07 | 0.08 | 0.358 | 0.23 | | | | | | | | |
| Socio-economic resilience | 0.09 | 0.1 | 0.35 | | | | | | | | | |
| Infrastructure adequacy | 0.22 | 0.07 | 0.002 | | 0.16 | 0.07 | **0.015** | | 0.24 | 0.08 | **0.003** | |
| Public health vulnerabilities | -0.05 | 0.15 | 0.715 | | | | | | | | | |
| **Region of the Americas (24 countries)** | | | | | | | | 0.81 | | | | 0.8 |
| **Prevention of the emergence Score** | | | | | | | | | | | | |
| Antimicrobial resistance (AMR) | 0.05 | 0.06 | 0.379 | 0.47 | | | | | | | | |
| Zoonotic disease | 0.18 | 0.09 | 0.066 | | | | | | | | | |
| Biosecurity | 0.08 | 0.05 | 0.105 | | | | | | | | | |
| Biosafety | 0.08 | 0.05 | 0.105 | | | | | | | | | |
| **Detection and reporting score** | | | | 0.7 | | | | | | | | |
| Laboratory systems strength and quality | 0.15 | 0.06 | 0.015 | | 0.2 | 0.06 | **0.003** | | 0.33 | 0.11 | **0.008** | |
| Laboratory supply chains | | | | | | | | | | | | |
| Real-time surveillance and Reporting | 0.14 | 0.06 | 0.036 | | 0.03 | 0.06 | 0.569 | | | | | |
| Surveillance data accessibility and transparency | 0.03 | 0.07 | 0.608 | | | | | | | | | |
| Case-based investigation | 0.08 | 0.09 | 0.353 | | | | | | | | | |
| **Respond rapid score** | | | | | | | | | | | | |
| Emergency preparedness and response planning | 0.25 | 0.09 | 0.014 | 0.41 | -0.05 | 0.08 | 0.502 | | | | | |
| Emergency response operation | 0.12 | 0.05 | 0.016 | | 0.21 | 0.08 | **0.017** | | 0.26 | 0.09 | **0.007** | |
| Health capacity in clinics, hospitals and community care centers | 0.12 | 0.1 | 0.246 | | | | | | | | | |
| **Health systems score** | | | | | | | | | | | | |
| Supply chain for health system and healthcare workers | 0.25 | 0.11 | 0.035 | 0.73 | 0.28 | 0.11 | **0.023** | | 0.42 | 0.21 | 0.065 | |
| Medical countermeasures and personnel deployment | <0.01 | 0.04 | 0.908 | | | | | | | | | |
| Capacity to test and approve new medical countermeasures | 0.18 | 0.07 | 0.015 | | 0.2 | 0.07 | **0.009** | | 0.23 | 0.11 | 0.058 | |
| **Risk environment score** | | | | | | | | | | | | |
| Infrastructure adequacy | | | | | 0.16 | 0.1 | 0.127 | | | | | |

*(Continued)*

**Table 5.** (Continued)

| Indicators by category | Model 1* | | | | Model 2** | | | | Model 3*** | | | |
|---|---|---|---|---|---|---|---|---|---|---|---|---|
| | Average† | SE | p-value | R2 | Average† | SE | p-value | R2 | Average† | SE | p-value | R2 |
| European Region (43 countries) | | | | | | | | 0.5 | | | | 0.33 |
| **Prevention of the emergence Score** | | | | | | | | | | | | |
| Antimicrobial resistance (AMR) | | | | | 0.28 | 0.13 | **0.035** | | 0.33 | 0.13 | **0.016** | |
| **Health systems score** | | | | | | | | | | | | |
| Capacity to test and approve new medical countermeasures | | | | | 0.03 | 0.14 | 0.826 | | | | | |
| **Risk environment score** | 0.95 | 0.35 | 0.01 | | | | | | | | | |
| Socio-economic resilience | 0.04 | 0.15 | 0.792 | 0.3 | | | | | | | | |
| Infrastructure adequacy | 0.24 | 0.1 | 0.028 | | 0.368 | 0.14 | **0.014** | | 0.39 | 0.15 | **0.012** | |
| Public health vulnerabilities | 0.2 | 0.14 | 0.157 | | | | | | | | | |
| Others regions (32 countries) | | | | | | | | 0.35 | | | | |
| **Respond rapid score** | | | | 0.31 | | | | | | | | |
| Risk communication | 0.02 | 0.01 | 0.019 | | 0.01 | <0.01 | 0.061 | | | | | |
| Trade and travel restrictions | 0.02 | 0.01 | 0.003 | | 0.01 | <0.01 | **0.048** | | | | | |
| **Health systems score** | | | | | | | | | | | | |
| Healthcare access | | | | | -0.006 | 0.01 | 0.324 | | | | | |
| **Risk environment score** | | | | | | | | | | | | |
| Political and security risk | | | | | 0.002 | 0.01 | 0.633 | | | | | |

*Adjusted for statistically significant indicators in each category in the crude analyses.

**Adjusted for the statistically significant indicators in model 1 and for the statistically significant indicators in the crude analyses.

*** Adjusted for all indicators that were statistically significant in model 2.

the GHS Index as such, since the indicators are based on publicly available documentation to quantify country capacity. Fourth, because of its design and the nature of our data at the country and regional level, our study is limited to presenting a correlation analysis and cannot prove causation. Finally, in estimating the Mpox case rate, we include no variables to adjust or standardize for the Mpox case rate within countries or regions, given the unavailability of country data on the WHO website. Failure to adjust for the Mpox case rate may have resulted in potential bias in country estimates.

Despite these limitations, the GHS Index is an important tool with a sound, rigorous/validated methodology for assessing global health security and pandemic preparedness. Data for each indicator and sub-indicator were obtained from WHO repositories. In addition, it is important to highlight that optimal GHS Index performance alone does not mean that health systems are prepared to respond to an Mpox outbreak. Therefore, future studies should evaluate the role of health systems, combined with the GHS Index, in assessing how to respond to an Mpox outbreak. According to the GHS Index Report, the overall scores summarize a country's capacity across all categories, but countries with high overall scores may have low scores at the category, indicator and sub-indicator levels [15]. Therefore, scores at the category, indicator and sub-indicator levels have a greater influence on outbreak-related outcomes than overall scores [15]. Because of this, an analysis of the GHS Index suggests that adjusted analyses of the correlation between country scores and health outcomes at the category, indicator and sub-indicator levels should be performed, as we did in the present study [15]. Finally, the GHS Index is not a predictive tool, and its conclusion remains that no country is adequately prepared to respond to epidemics or pandemics. Therefore, our results are limited to

explaining whether countries with high GHS Index category and indicator levels have high Mpox rates.

The results of this study show that nonlinear relationships can be more complex than linear ones. For example, Finland, which has a high score in the GHS Index categories, has a low rate of Mpox cases, whereas Peru has an average score in the GHS Index categories and a high rate of Mpox cases. In addition, in the overall analysis, there was a high dispersion of the correlation coefficient between countries in Europe and the Americas. However, in the adjusted analysis by category and indicator, some variables were associated with the Mpox case rate. Adjusted analysis allows the identification of indicators explaining correlations, but sometimes the correlation may be influenced by variables not measured in the study. Finally, the R2 coefficient used to assess the proportion of variability in the dependent variable confirms that the model is better fitted for the countries of the region of the Americas.

## 5. Conclusions

In summary, countries with higher levels of emergency preparedness indicators on the GHS index (risk environment, AMR, risk communication, JEE and PVS, and adequacy of infrastructure) had higher Mpox rates. In the region of the Americas, countries with higher scores on indicators of laboratory systems strength and quality, and emergency response operation had higher Mpox rates. In the European region, countries with higher scores on indicators of antimicrobial resistance and infrastructure adequacy had higher Mpox rates. Finally, in other regions, countries with higher scores on indicators of trade and travel restrictions had a higher Mpox rates. These findings suggest that countries with higher health security capacity on the GHS index are better equipped to detect more cases of Mpox in the current outbreak. Therefore, a robust health infrastructure and preventive measures to ensure an accessible health system are critical to detecting Mpox cases. To improve our understanding of the role of capacities in preparing for and responding to an outbreak, countries should continue to measure health security capacities using the GHS Index.

## Supporting information

**S1 Table. STROBE statement—Checklist of items that should be included in reports of cross-sectional studies.**
(DOC)

**S2 Table. GHS Index category indicator variables selected for the study.**
(DOCX)

## Author Contributions

**Conceptualization:** Max Carlos Ramírez-Soto.

**Data curation:** Max Carlos Ramírez-Soto, Hugo Arroyo-Hernández.

**Formal analysis:** Max Carlos Ramírez-Soto, Hugo Arroyo-Hernández.

**Investigation:** Max Carlos Ramírez-Soto, Hugo Arroyo-Hernández.

**Methodology:** Max Carlos Ramírez-Soto, Hugo Arroyo-Hernández.

**Project administration:** Max Carlos Ramírez-Soto.

**Software:** Max Carlos Ramírez-Soto, Hugo Arroyo-Hernández.

**Validation:** Max Carlos Ramírez-Soto, Hugo Arroyo-Hernández.

**Visualization:** Max Carlos Ramírez-Soto.

**Writing – original draft:** Max Carlos Ramírez-Soto, Hugo Arroyo-Hernández.

**Writing – review & editing:** Max Carlos Ramírez-Soto, Hugo Arroyo-Hernández.

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
