## [Decision Letter · Decision Letter 0]

24 Jun 2024

PONE-D-24-19347High Global Health Security Index is a determinant of early detection and response to monkeypox: A cross-sectional studyPLOS ONE

Dear Dr. Ramírez-Soto,

Thank you for submitting your manuscript to PLOS ONE. After careful consideration, we feel that it has merit but does not fully meet PLOS ONE’s publication criteria as it currently stands. Therefore, we invite you to submit a revised version of the manuscript that addresses the points raised during the review process.

We look forward to receiving your revised manuscript.

Kind regards,

Moises Leon Juarez

Academic Editor

PLOS ONE

Journal Requirements:

3. We note that Figure S1 in your submission contain map images which may be copyrighted. All PLOS content is published under the Creative Commons Attribution License (CC BY 4.0), which means that the manuscript, images, and Supporting Information files will be freely available online, and any third party is permitted to access, download, copy, distribute, and use these materials in any way, even commercially, with proper attribution. For these reasons, we cannot publish previously copyrighted maps or satellite images created using proprietary data, such as Google software (Google Maps, Street View, and Earth). For more information, see our copyright guidelines: http://journals.plos.org/plosone/s/licenses-and-copyright.

We require you to either present written permission from the copyright holder to publish these figures specifically under the CC BY 4.0 license, or remove the figures from your submission:

a. You may seek permission from the original copyright holder of Figure S1 to publish the content specifically under the CC BY 4.0 license.  

Additional Editor Comments:

We inform you that your work has been reviewed by the reviewers. Based on their comments and observations, we consider that you need to make rigorous changes for its possible publication in a journal. Below, I send you some suggestions that should be considered according to the reviewers' comments.

Reviewer 1

It is a very well-documented and stated paper. However, a relevant previous paper proposes a relationship between monkeypox confirmed cases and deaths in a very similar period as the chosen by the one authors of this work (May 1, 2022, to May 15, 20,23). The authors of this paper found that the Pearson correlation test showed a significant positive linear relationship between total cases and total deaths with the GHSI. It is advisable to cite the direct aforegoing paper, which is considered to establish these relationships. Hasan MN, ArunSundar MS, Bhattacharya P, Islam A. Exploring the relationship between the Global Health Security Index and monkeypox: an analysis of preparedness and response capacities. International Journal of Surgery: Global Health. 2023 Jul 1;6(4). https://doi.org/10.1097/GH9.0000000000000229

Reviewer 2

The manuscript by Ramirez-Soto and Arroyo Hernandez presents an interesting analysis of the correlations between the GHS index and Mpox cases in different countries. Although the study is intriguing, I have major concerns.

1. What were the criteria for normalization to a scale of 0 to 100?

2. In Table 1, does “RIQ” stand for “IQR” in Spanish?

3. Does the international abbreviation stand for “until”?

4. Provide a more detailed description of each variable considered for correlation, especially AMR.

5. Describe variables that might differ regarding health safety capacities in the GHS index, specifically those implemented for COVID-19 versus MPXV.

6. Improve the quality of the figures.

7. Another study has evaluated the GHS index and Mpox preparedness. The authors should justify the differences and the novelty of this study compared to the previous one. Compare the results with other analyses published in articles like DOI: 10.1097/GH9.0000000000000229 and earlier reports by the authors in Peru.

8. The authors should clearly state the limitations in the conclusions. The claim that countries with a high GHS index can successfully respond to Mpox outbreaks is not sufficiently supported by the correlation data presented. More evidence is needed to confirm this claim.

Reviewers' comments:

Reviewer's Responses to Questions

**Comments to the Author**

1. Is the manuscript technically sound, and do the data support the conclusions?

Reviewer #1: Yes

Reviewer #2: Partly

2. Has the statistical analysis been performed appropriately and rigorously? 

Reviewer #1: Yes

Reviewer #2: I Don't Know

3. Have the authors made all data underlying the findings in their manuscript fully available?

Reviewer #1: Yes

Reviewer #2: Yes

4. Is the manuscript presented in an intelligible fashion and written in standard English?

Reviewer #1: Yes

Reviewer #2: Yes

5. Review Comments to the Author

**Reviewer #1:** It is a very well documented and stated paper. However, there is a relevant previous paper which propose to establish a relation between total monkeypox confirmed cases and deaths in a very similar period of time as the chosen by the authors of this work (May 1, 2022 to May 15, 2023). Authors of this paper found that Pearson correlation test showed a significant positive linear relationship between total cases and total deaths with the GHSI. It should be highly advisable to cite maybe the direct aforegoing paper which considered to establish these relationships. Hasan MN, ArunSundar MS, Bhattacharya P, Islam A. Exploring the relationship between the Global Health Security Index and monkeypox: an analysis of preparedness and response capacities. International Journal of Surgery: Global Health. 2023 Jul 1;6(4). https://doi.org/10.1097/GH9.0000000000000229

**Reviewer #2: **The manuscript by Ramirez-Soto and Arroyo Hernandez presents an interesting analysis of the correlations between the GHS index and Mpox cases in different countries. Although the study is intriguing, I have major concerns.

1. What were the criteria for normalization to a scale of 0 to 100?

2. In Table 1, does “RIQ” stand for “IQR” in Spanish?

3. Does the international abbreviation stand for “until”?

4. Provide a more detailed description of each variable considered for correlation, especially AMR.

5. Describe variables that might differ regarding health safety capacities in the GHS index, specifically those implemented for COVID-19 versus MPXV.

6. Improve the quality of the figures.

7. Another study has evaluated the GHS index and Mpox preparedness. The authors should justify the differences and the novelty of this study compared to the previous one. Compare the results with other analyses published in articles like DOI: 10.1097/GH9.0000000000000229 and previous reports by the authors in Peru.

8. The authors should clearly state the limitations in the conclusions. The claim that countries with a high GHS index can successfully respond to Mpox outbreaks is not sufficiently supported by the correlation data presented. More evidence is needed to confirm this claim.

6. PLOS authors have the option to publish the peer review history of their article (what does this mean?). If published, this will include your full peer review and any attached files.

Reviewer #1: **Yes: **Jorge Alberto Álvarez Díaz

Reviewer #2: No

---

## [Author Response · Author response to Decision Letter 0]

4 Jul 2024

Response to reviewers 

Title of the article: High Global Health Security Index is a determinant of early detection and response to monkeypox: A cross-sectional study

Reference number: PONE-D-24-19347

Journal Requirements:

Response: Thank you for your comments.

Response: Thank you for your comments.

3. We note that Figure S1 in your submission contain map images which may be copyrighted. All PLOS content is published under the Creative Commons Attribution License (CC BY 4.0), which means that the manuscript, images, and Supporting Information files will be freely available online, and any third party is permitted to access, download, copy, distribute, and use these materials in any way, even commercially, with proper attribution. For these reasons, we cannot publish previously copyrighted maps or satellite images created using proprietary data, such as Google software (Google Maps, Street View, and Earth). For more information, see our copyright guidelines: http://journals.plos.org/plosone/s/licenses-and-copyright.

We require you to either present written permission from the copyright holder to publish these figures specifically under the CC BY 4.0 license, or remove the figures from your submission:

Response: Thank you for your comments. Figure S1 has been removed. 

Response: Thank you for your comments. The references have been corrected.

Reviewer #1: 

It is a very well-documented and stated paper. However, a relevant previous paper proposes a relationship between monkeypox confirmed cases and deaths in a very similar period as the chosen by the one authors of this work (May 1, 2022, to May 15, 20,23). The authors of this paper found that the Pearson correlation test showed a significant positive linear relationship between total cases and total deaths with the GHSI. It is advisable to cite the direct aforegoing paper, which is considered to establish these relationships. Hasan MN, ArunSundar MS, Bhattacharya P, Islam A. Exploring the relationship between the Global Health Security Index and monkeypox: an analysis of preparedness and response capacities. International Journal of Surgery: Global Health. 2023 Jul 1;6(4). https://doi.org/10.1097/GH9.0000000000000229.

Response: Thank you for your comments. According Avi et al. The value proposition of the Global Health Security Index. BMJ Glob Health. 2020;5(10):e003648. Drawing simple correlations with countries’ overall scores does not account for the fact that these scores are meant to capture capacities spanning the breadth of the health security life cycle, from outbreak prevention at the source, to early detection, to rapid response. In other words, overall scores summarise country capacities across all categories. However, countries with high overall scores may still have low category-level, indicator-level and sub-indicator-level scores that more strongly influence outbreak-associated outcomes. Therefore, we encourage users wishing to analyse correlations between country scores and health outcomes to examine scores at more granular levels, adjust model weights to reflect country contexts and priorities and consider more nuanced outcomes when analysing countries’ performances during outbreaks. In these context, studies of Hasan et al. International Journal of Surgery: Global Health. 2023 Jul 1;6(4), and Ramírez-Soto MC. Monkeypox Outbreak in Peru. Medicina (Kaunas). 2023 Jun 6;59(6):1096, They only perform a correlation between the overall GHS index score and the Mpox case, and cannot represent the category, indicator and sub-indicator of the GHS index.

Therefore, we hypothesized that the GHS indicators of detection, prevention, reporting, health system, rapid response, compliance with international standards and risk environment could explain global and regional Mpox case rates (category and sub-indicators). Thus, we aimed to analyze the correlation between the GHS index and Mpox case rates in different countries and regions. The evaluation of the GHS on Mpox is critical for transmission control and will provide important insights into the relationships between the GHS index and Mpox cases rates. In addition, in Discussion section, Limitations and strengths, We explain as a strength of the study the analysis of indicators and sub-indicators “According to the GHS Index Report, the overall scores summarize a country's capacity across all categories, but countries with high overall scores may have low scores at the category, indicator and sub-indicator levels. Therefore, scores at the category, indicator and sub-indicator levels have a greater influence on outbreak-related outcomes than overall scores. Because of this, an analysis of the GHS Index suggests that adjusted analyses of the correlation between country scores and health outcomes at the category, indicator and sub-indicator levels should be performed, as we did in the present study”.

We have also included a paragraph in the Introduction section on the limitations of the GHS index overall scores.

Reviewer #2: 

The manuscript by Ramirez-Soto and Arroyo Hernandez presents an interesting analysis of the correlations between the GHS index and Mpox cases in different countries. Although the study is intriguing, I have major concerns.

Comment 1. What were the criteria for normalization to a scale of 0 to 100?

Response: Thank you for your comments. We have explained these details in paragraph 2 of the Methods section “The overall score (0 to 100) for each country is a weighted sum of the scores in the six categories. The score for each category is given on a scale from 0 to 100. A score of 100 represents the most favourable health security conditions, but does not mean that a country has a perfect national health security situation. A score of 0 is the least favourable condition, but does not imply a lack of capacity. Each category is standardized on the basis of the sum of its underlying indicators and sub-indicators, and a weight is assigned to each. The weights in the model are dynamic and can be changed by the users. To facilitate reproducible comparisons across countries, categories and indicators, measurements have been normalized to a scale of 0 to 100”.

Comment 2. In Table 1, does “RIQ” stand for “IQR” in Spanish?

Response: Thank you for your comments. The sentences in Results section has been corrected. 

Comment 3. Does the international abbreviation stand for “until”?

Response: Thank you for your comments. In the manuscript we have not used the abbreviation “until”. 

Comment 4. Provide a more detailed description of each variable considered for correlation, especially AMR. 

Response: Thank you for your comments. In Table S2 is shown GHS Index indicator and sub-indicators variables selected for the study. 

Comment 5. Describe variables that might differ regarding health safety capacities in the GHS index, specifically those implemented for COVID-19 versus MPXV.

Response: Thank you for your comments. We included the GHS index categories associated with COVID-19 mortality. We did not find any other studies evaluating GHS index categories and indicators with COVID-19 cases. In the case of Mpox, the two previous studies only evaluated the overall GHS index score, which is not representative of the categories and indicators and has limitations. To date, there have been no further analyses of the GHS index categories and indicators for monkeypox case rates. This is explained in paragraph 2 of the introduction. 

Comment 6. Improve the quality of the figures.

Response: Thank you for your comments. The figures attached in the PDF are shown with high resolution.

Comment 7. Another study has evaluated the GHS index and Mpox preparedness. The authors should justify the differences and the novelty of this study compared to the previous one. Compare the results with other analyses published in articles like DOI: 10.1097/GH9.0000000000000229 and earlier reports by the authors in Peru.

Response: Thank you for your comments. According Avi et al. The value proposition of the Global Health Security Index. BMJ Glob Health. 2020;5(10):e003648. Drawing simple correlations with countries’ overall scores does not account for the fact that these scores are meant to capture capacities spanning the breadth of the health security life cycle, from outbreak prevention at the source, to early detection, to rapid response. In other words, overall scores summarise country capacities across all categories. However, countries with high overall scores may still have low category-level, indicator-level and sub-indicator-level scores that more strongly influence outbreak-associated outcomes. Therefore, we encourage users wishing to analyse correlations between country scores and health outcomes to examine scores at more granular levels, adjust model weights to reflect country contexts and priorities and consider more nuanced outcomes when analysing countries’ performances during outbreaks. In these context, studies of Hasan et al. International Journal of Surgery: Global Health. 2023 Jul 1;6(4), and Ramírez-Soto MC. Monkeypox Outbreak in Peru. Medicina (Kaunas). 2023 Jun 6;59(6):1096, They only perform a correlation between the overall GHS index score and the Mpox case, and cannot represent the category, indicator and sub-indicator of the GHS index.

Therefore, we hypothesized that the GHS indicators of detection, prevention, reporting, health system, rapid response, compliance with international standards and risk environment could explain global and regional Mpox case rates (category and sub-indicators). Thus, we aimed to analyze the correlation between the GHS index and Mpox case rates in different countries and regions. The evaluation of the GHS on Mpox is critical for transmission control and will provide important insights into the relationships between the GHS index and Mpox cases rates. In addition, in Discussion section, Limitations and strengths, We explain as a strength of the study the analysis of indicators and sub-indicators “According to the GHS Index Report, the overall scores summarize a country's capacity across all categories, but countries with high overall scores may have low scores at the category, indicator and sub-indicator levels. Therefore, scores at the category, indicator and sub-indicator levels have a greater influence on outbreak-related outcomes than overall scores. Because of this, an analysis of the GHS Index suggests that adjusted analyses of the correlation between country scores and health outcomes at the category, indicator and sub-indicator levels should be performed, as we did in the present study”.

We have also included a paragraph in the Introduction section on the limitations of the GHS index overall scores.

Comment 8. The authors should clearly state the limitations in the conclusions. The claim that countries with a high GHS index can successfully respond to Mpox outbreaks is not sufficiently supported by the correlation data presented. More evidence is needed to confirm this claim.

Response: Thank you for your comments. Conclusion has been corrected “These finding suggest that countries with higher health security capacity on the GHS index are better equipped to detect more cases of Mpox in the current outbreak. Therefore, a robust health infrastructure and preventive measures to ensure an accessible health system are critical to detecting Mpox cases. To improve our understanding of the role of capacities in preparing for and responding to an outbreak, countries should continue to measure health security capacities using the GHS Index”.

---

## [Editor Report · Decision Letter 1]

8 Jul 2024

High Global Health Security Index is a determinant of early detection and response to monkeypox: A cross-sectional study

PONE-D-24-19347R1

Dear Dr. Ramírez-Soto,

We’re pleased to inform you that your manuscript has been judged scientifically suitable for publication and will be formally accepted for publication once it meets all outstanding technical requirements.

Kind regards,

Moises Leon Juarez

Academic Editor

PLOS ONE

Additional Editor Comments (optional):

Dear Dr. Ramírez Soto

It is a pleasure to inform you about the decision made about his work: High Global Health Security Index is a determinant of aerial detection and response to monkeypox: A cross-sectional study". Based on the changes and recommendations that the reviewers made to your work, we have decided to accept your work for publication.
---

## [Editor Report · Acceptance letter]

17 Jul 2024

PONE-D-24-19347R1 

PLOS ONE

Dear Dr. Ramírez-Soto, 

I'm pleased to inform you that your manuscript has been deemed suitable for publication in PLOS ONE. Congratulations! Your manuscript is now being handed over to our production team.

Kind regards, 

on behalf of

Dr. Moises Leon Juarez 

Academic Editor

PLOS ONE